# Effects of Dried Onion Powder and Quercetin on Obesity-Associated Hepatic Menifestation and Retinopathy

**DOI:** 10.3390/ijms231911091

**Published:** 2022-09-21

**Authors:** Wen-Lung Chang, Pei-Yi Liu, Shu-Lan Yeh, Huei-Jane Lee

**Affiliations:** 1Institute of Medicine, Chung Shan Medical University, Taichung 40221, Taiwan; 2Department of Nutritional Science, Chung Shan Medical University, Taichung 40221, Taiwan; 3Department of Nutrition, Chung Shan Medical University Hospital, Taichung 40221, Taiwan; 4Department of Biochemistry, School of Medicine, Chung Shan Medical University, Taichung 40221, Taiwan; 5Department of Clinical Laboratory, Chung Shan Medical University Hospital, Taichung 40201, Taiwan

**Keywords:** dried onion powder, quercetin, high-fat diet, obesity, retinopathy

## Abstract

Onion (*Allium cepa* L.), rich in flavonoids (particularly quercetin), reportedly has anti-obesity properties, but the underlying mechanisms and associated health issues remain unclear. In this study, we compared the effects of dried onion powder (DO) with that of quercetin on high-fat diet (HFD)-induced obesity, nonalcoholic fatty liver disease, and retinal neovascularization. Briefly, rats (*n* = 9–10 per group) were divided into control, HFD alone (43% fat), HFD + DO (1% DO), HFD + 5DO (5% DO, *w*/*w*), and HFD + quercetin (180 mg/kg). After 12 weeks, body fat, markers of metabolism, fatty liver, steatohepatitis, and retinopathy were analyzed. The results revealed that DO and 5DO dose-dependently suppressed body weight, visceral and subcutaneous fat accumulation, and epididymal adipocyte in HFD-fed rats. DO also decreased HFD-induced ALT, AST, free fatty acid, glucose, proinflammatory cytokines, and oxidative stress. DO and 5DO groups had lower triglycerides, total cholesterol, proinflammatory cytokine levels, and ACC-α (a fatty acid synthesis–associated enzyme) expression but higher hepatic antioxidant enzyme activities and fecal lipids. 5DO exhibited better or similar efficacy to quercetin. Both 5DO and quercetin increased fecal levels of acetic acid and butyric acid similarly. They also reduced lipid peroxidation of the eye, retinal adiposity, and neovascularization. However, quercetin resulted in a more apparent decrease in regulation of the Raf/MAPK pathway than DO in eye specimens. Conclusively, DO suppresses visceral, subcutaneous, and liver fat accumulation better than quercetin likely due to higher fecal fat excretion and lower oxidative stress, proinflammatory cytokine levels, and ACC-α expression. Quercetin regulating signal pathways is better than DO at reducing retinal adiposity and neovascularization.

## 1. Introduction

Dietary habits and lifestyle influence the prevalence of overweight and obesity [1]; the health burden of these conditions is increasing worldwide, including in Taiwan, and they are causally linked with the development of several chronic diseases such as diabetes, cardiovascular disease, cancer, and nonalcoholic fatty liver disease (NAFLD) [2]. Overweight and obesity display abnormal or excessive fat accumulation, including visceral and subcutaneous fat [3]. Visceral fat has higher fat synthesis and degradation activity than subcutaneous fat. Free fatty acids and adipokines/cytokines released from visceral fat and flowing into the liver contribute to the development of NAFLD [4] and metabolic disorders. A cross-sectional human study indicated that visceral fat is directly associated with liver inflammation and fibrosis, independent of insulin resistance and hepatic steatosis [5]. However, compared with subcutaneous fat, visceral adipose tissue is more severely infiltrated by macrophages [6]; hypertrophic adipocytes secrete proinflammatory cytokines, such as interleukin (IL)-6 and tumor necrosis factor (TNF)-α, into circulation and increase inflammation in various organs [7]. A prospective cross-sectional single-visit study revealed that a high-percentage body fat is associated with dry eye [8]. An analysis from the UK Biobank indicated that the microvascular characteristics of the retina are strongly associated with adiposity [9]. Thus, visceral fat reduction may be a target for improving NAFLD and eye diseases caused by metabolic dysfunction.

Onion (*Allium cepa* L.) is a vegetable commonly consumed worldwide and contains diverse phytochemicals, including organosulfur compounds, phenolic compounds, polysaccharides, and saponins [10]. Onion and its bioactive compounds have various pro-health functions, including anti-inflammatory, antioxidant, and anti-obesity properties [10]. Onion prevented oxidative stress by enhancing the activity of antioxidant enzymes-such as superoxide dismutase (SOD), catalase, and glutathione peroxidase (GPx) in hypercholesterolemic rats [11].

A diet containing 5% onion (*w*/*w*) was reported to decrease weight gain, hepatic total cholesterol level, and plasma glutamate oxaloacetate transaminase level in Sprague Dawley rats [12]. In addition, 7% onion powder supplementation prevented high-fat, high-sugar diet-induced NAFLD, as evidenced by decreasing hepatic steatosis, hepatic TNF expression, and plasma alanine transaminase (ALT) levels in rats [13]. The beneficial effects of onion in NAFLD may be associated with the regulation of its risk factors, such as hyperlipidemia, oxidative stress, hyperglycemia, and inflammation [14]. However, a clinical trial concluded that daily intake of 9 g heated onion powder (sterilized at 60 °C for 120 min) for 12 weeks did not affect visceral fat area (VFA), except for participants with lower levels of high-density lipoprotein cholesterol (HDL-C) [15]. As mentioned above, visceral fat accumulation is an important factor contributing to the diseases caused by obesity, thus more studies are necessary to confirm whether onion powder regulates visceral fat accumulation.

Therefore, the major aims of this study were to investigate the effects of freeze-dried onion powder (DO) on a high-fat diet (HFD)-induced disorder in VFA accumulation, liver health, and retinal neovascularization in rats and explore the possible mechanisms. As DO has reportedly shown that the contents of flavonoid (90% are quercetin and quercetin derivatives) in black onion and the indicated flavonoids are decreased during heat treatment [16], we used DO, which was prepared by freeze-drying at −40 °C, to perform the current study. Additionally, DO is a kind of condensed bioactive powder, which reduces the required amount of consumption for the health benefits of onion and may facilitate the development of onion as a functional food for preventing obesity and the associated health issues. Moreover, it has been indicated that quercetin, a major flavonoid present in onions [17] can protect mice against HFD-induced NAFLD by modulating intestinal microbiota imbalance and related gut–liver axis activation [18]. However, whether those co-existing compounds, such as fibers, in onion affect such effects of quercetin is unclear, thus we also compared the effects of DO with those of quercetin alone.

## 2. Results

### 2.1. Food Intake, Body Weight, Visceral (Epididymal, Perirenal, and Mesenteric) and Subcutaneous Fat Weight

The rats at average body weight 215 ± 8 g were randomly divided into control diet (*n* = 9); HFD (*n* = 10); HFD + DO (*n* = 10); HFD + 5DO (*n* = 10); HFD + Q (*n* = 9). After being fed the indicated diets for 12 weeks, compared with the control rats, the HFD-fed rats had significantly higher energy intake, body weight, and fat weight but significantly lower food intake mass (Table 1). DO and quercetin did not significantly reduce the energy intake in the HFD-fed rats but tended to decrease body weight, especially 5DO. Furthermore, compared with the HFD group, both the DO and 5DO groups had significantly lower epididymal, perirenal, and mesenteric fat tissue mass (16–27% and 18–42% lower, respectively); 5DO did significantly decrease subcutaneous fat by 18%, however (Table 2). Quercetin did not significantly decrease visceral or subcutaneous fat weight. Histopathology of epididymal adipose tissue revealed that DO decreased adipocyte size in the HFD-fed rats (Figure 1). A difference in efficacy between 5DO and Q was found; it showed 5DO possessed better effects in lowering body weight and fat than that of quercetin.

### 2.2. Effects of DO and Quercetin on Blood Biochemistry Values

After 12-week treatment, HFD significantly increased the levels of aspartate transaminase (AST), alanine transaminase (ALT), interleukin (IL)-6, tumor necrosis factor (TNF)-α, and oxidative stress (TBARs) and decreased the HDL-C level (Table 3). It revealed that HFD impaired the liver function, increased inflammatory cytokines and oxidative stress. DO not only decreased all these parameters but also decreased serum free fatty acid and glucose levels in a dose-dependent manner in the HFD-fed rats. In addition, HFD tended to decrease serum triglyceride, total cholesterol, and LDL-C levels; 5DO resulted in significantly lower triglyceride and LDL-C levels than were discovered in the HFD alone group. The effects of quercetin on lowering the aforementioned HFD-increased blood biochemistry values were weaker or similar to those of 5DO.

### 2.3. Effects of DO and Quercetin on Liver Histology and Lipid Accumulation

HFD significantly increased the levels of triglycerides, total cholesterol, IL-6, and TBARs but decreased glutathione peroxidase activity and glutathione levels in the liver (Table 4). The effects of HFD were significantly lower in HFD + DO and HFD + 5DO groups. Compared with the HFD group, the HFD + 5OD also had significantly lower TNF-α concentration and higher SOD and catalase activities. Consistently, HE-stained images of the liver (Figure 2A) revealed that the HFD induced histopathological changes in the fatty liver by tending to increase microvascular and macrovascular steatosis as well as inflammation (Table 5). Furthermore, 5DO, but not DO, significantly reduced the damage scores for damage induced by the HDF. DO also decreased the HFD-induced mRNA and protein expression of ACC-α (Figure 2B,C). Except for ACC-α expression, the reverse effects of quercetin on all aforementioned HFD-induced changes were weaker than those observed in the HFD + 5DO group.

### 2.4. Effects of DO and Quercetin on Lipid Profiles in Feces

Compared with the control rats, the HFD-fed rats had significantly higher triglyceride and total cholesterol levels in feces but comparable short-chain fatty acid content in feces. DO, especially 5DO, further increased the levels of total lipids, triglycerides, and cholesterol. Compared with the control group, the HFD + 5DO had significantly higher acetic acid and butyric acid levels. Quercetin did not affect the fecal levels of total lipid, triglycerides, and cholesterol in the HFD-fed rats but significantly increased their acetic acid and butyric acid levels (Table 6).

### 2.5. Effects of DO and Quercetin on Retinopathy

Our previous study indicated that lipids accumulate within the choroid layer of the retina in HFD-fed animals [19]. To show lipid deposits in the retina, eye specimens were examined. As presented in Figure 3A, the histological results indicated that lipids were deposited within the choroid layer mainly in the HFD groups. DO and quercetin had observable beneficial effects. To further reveal the level of neovascularization, the sections were stained with an endothelial cell marker, CD31. Neovascularization was increased in the HFD group and decreased in the HFD + DO and HFD + quercetin groups (Figure 3B).

### 2.6. Effects of DO and Quercetin on Lipid Peroxidation and the PI3K/Akt and Ras/Raf/MEK Pathways in Eye Specimens

To explore the level of oxidized lipids accumulated in the eye, lipid peroxidation was detected in eye lysate. As presented in Figure 4A, the degree of lipid peroxidation was significantly higher (37% higher) in the eye of the HFD group compared with the control but DO and quercetin mitigated 86% and 49% of this increase in the amount of lipid peroxidation, respectively. 5DO but the groups receiving DO and quercetin had 86% and 49% lower lipid peroxidation than the control group, respectively. The mechanisms of retinopathy involve regulation of the PI3K/Akt and Ras/Raf/MEK pathways. The results presented in Figure 4B show that compared with the control group, the HFD group had PI3K levels. Quercetin decreased the PI3K level in the HFD-fed rats. Ras/Raf/MEK levels, but not Akt and p-Akt levels, were significantly higher in the HFD group than the control group. DO significantly reduced Ras levels, and quercetin reduced c-Raf and MEK levels. The level of p-ERKs in the HFD group was the same as that in the control group (Figure 4C).

## 3. Discussion

Our findings indicate that DO can significantly reduce HFD-induced body weight gain, body fat, and hepatic lipid accumulation and that 5DO was more efficacious than quercetin. The quercetin content of fresh onions was reported to be approximately 284–486 mg/kg [20]. Additionally, the moisture content of fresh onions ranged from 84% to 91% [21]; thus, the quercetin content in our dried onion was approximately 250 mg/100 g dried onion, which was comparable to previous findings. Jung et al. revealed that in a 9-week feeding intervention, 0.025% (*w*/*w*) quercetin reduced HFD-induced fat accumulation in the liver of C57/B6 mice [22]. In our current study, the quercetin content in the diet was 180 mg/kg, which is equal to the quercetin content from 5% DO (5DO). Compared with 5DO, quercetin supplementation had lower efficacy in reducing body weight and lipid accumulation, implying differences in the mechanisms through which DO and quercetin exert their effects in HFD-fed rats. We presumed that multiple mechanisms contributed to the inhibiting effects of DO on HFD-induced fatty liver and fat accumulation, especially visceral fat.

First, DO increased fecal fat and short-chain fatty acid excretion, contributing to reductions in lipid content and weight. Moreover, increased fecal fat excretion decreased fat absorption and, thus, fat accumulation. Our data (Table 6) demonstrated that OD increased the level of total lipids, triglycerides, and cholesterol in feces in a dose-dependent manner, but quercetin had no effect. We speculated that the soluble fiber (such as fructans that is, inulin and fructooligosaccharides) in DO/5DO may contribute to the increased fecal lipid excretion, because insoluble fiber was supplemented in the diets in each group (Table 7). Hiel et al. [23] demonstrated that inulin supplementation improves postprandial hypertriglyceridemia by modulating gene expression and increases fecal lipid excretion. Soluble fibers, also known as prebiotics, stimulate the growth of probiotics and production of short-chain fatty acids [24]. Similar to 5DO, quercetin increased the levels of acetic acid and butyric acid, implying that quercetin has prebiotic capability. This finding agrees with the study of Porras et al. [18]. Probiotics and their derived short-chain fatty acids improve HFD-induced NAFLD by attenuating oxidative stress and inflammation [25].

Second, DO was found to have anti-inflammatory and antioxidant properties; OD and 5OD dose-dependently decreased the levels of proinflammatory cytokines, IL-6, TNF-α, oxidative stress markers, TBARS, in the serum and liver. Furthermore, DO increased hepatic antioxidant enzyme activity and glutathione levels. These effects could be partly attributed to the presence of quercetin, a well-known anti-inflammatory and antioxidative flavonoid [26]. Comparing the effects of DO with those of quercetin, we observed that the inhibiting effects of DO on these parameters were stronger than those of quercetin, implying that other factors present in onion provide additive or synergistic effects with quercetin. However, DO and quercetin similarly decreased the expression of HFD-induced ACC-α (Figure 2), which is a committed enzyme for de novo fatty acid synthesis in the liver and white adipose tissue [27], indicating that quercetin plays an important role in this mechanism.

Increasing adiposity has been associated with eye diseases such as microvascular characteristics of the retina [9] and age-related eye diseases [28]. In the current study, HFD-fed rats exhibited lipid deposition and neovascularization in the retinal choroid, both of which were attenuated in the HFD-fed rats given DO or quercetin supplement (Figure 3A,B). HFD-induced eye disease has been associated with oxidative stress [29]. Our results revealed that DO and quercetin reduced the oxidative marker TBARS, as they did in the serum and liver (Figure 4A). Oxidized lipids directly or indirectly form drusen that are deposited in the Bruch membrane of the retina and cause retinal pigment epithelial cell death [30]. Furthermore, oxidative stress triggers the upregulation of vascular endothelial growth factor to cause retinal neovascularization; it involves various cellular regulations, such as those of the PI3K/AKT protein kinase B pathway and the MAPK signaling pathway [31]. Herein, the level of PI3K was higher in the eye specimens of the HFD-fed rats than in those of the control rats; quercetin but not DO reduced the PI3K level (Figure 4B). Although the downstream protein AKT was not expressed significantly, PI3K may regulate mTOR phosphorylation to activate 4E-BP1 and affect angiogenesis [32]. Nonetheless, the correlations between HFD and retinal angiogenesis remain to be further addressed.

Our data indicated that the protein expression of Ras, Raf, MEK, and PI3K was 2.45-fold, 2.21-fold, 2.63-fold, and 3.03-fold higher, respectively, in the eye specimens of the HFD group than those of the control group. DO reduced the protein levels of Ras and MEK while quercetin reduced the levels of Raf, MEK, phosphor-MEK, and PI3K. Compared with those of quercetin, the protective effects of DO on HFD-induced fatty liver and fat accumulation were superior but on eye problems were poorer. This may be consequent to the net concentration within serum or organ quercetin. Quercetin is present in onions as various glycosides [33]. We prepared DO without heating, so its chain-breaking activity could not be elevated and the effect of quercetin could not be completely represented [34]. Therefore, in addition to quercetin, the effects of onion in reducing body lipids or liver fat, increasing fecal lipid excretion, increasing antioxidant effects, and reducing inflammatory cytokines may also be contributed by other active compounds in onion, such as soluble fibers. In the HFD + Q group, we used quercetin aglycone, which is absorbed more efficiently than quercetin glycoside [35]. Additionally, orally administered quercetin reportedly crosses the blood–brain barrier and accumulates in the brain, thus having strong antioxidant effects [36]. In our study, quercetin may have penetrated the blood–retina barrier to provide antioxidant benefits to the retina (Figure 4A) and further regulate protein expression, which may explain our results. Future studies should elucidate the mechanisms underlying quercetin transport into the retina.

In conclusion, DO and quercetin individually have benefits for reducing body lipids, liver fat, oxidative stress, and inflammatory cytokines. We explored DO or quercetin reduced retinal adiposity and neovascularization (Figure 5). The results revealed that DO or quercetin not only reduce lipid accumulation but also protect from HFD-induced retinal damage.

## 4. Materials and Methods

### 4.1. Preparation of Freeze-Dried Onion Powder

Fresh onion were obtained from Checheng Farmers’ Association (Pingtung County, Taiwan). After peeling off their skins, the onions were cut into small pieces. Next, the onion pieces were lyophilized at −40 °C for 24 hrs (Chaosu Company, Minhsiung Township, Chiayi County, Taiwan) then mechanically gridded into fine powder, and stored at −20 °C until use. The quercetin content of the DO was determined using high performance liquid chromatography [16]; 100 g of the DO contained 250 ± 3 mg of quercetin. DO insoluble fiber content was determined as described previously [37] and total fructan content was determined by AOAC 999.03-2005; 100 g of the DO contained 8.86 g crude fiber (insoluble fiber) and 3.23 g fructans (soluble fiber).

### 4.2. Animal Studies

All animal experiments were approved by the Institutional Animal Care and Use Committee at Chung Shan Medical University (Approval no. 2197). Animal care followed the International Guiding Principles for Biomedical Research Involving Animals [38]. Four-week-old male Sprague Dawley rats were obtained from BioLASCO Taiwan (YiLan County, Taiwan) and housed under laboratory conditions (18–23 °C, humidity: 55–60%, 12-h light/dark cycle). After being acclimated for 1 week (average body weight: 215 ± 8 g), the rats were randomly assigned to the following diet groups: control diet (*n* = 9), HFD (*n* = 10), HFD + DO (*n* = 10), HFD + 5DO (*n* = 10), and HFD + Q (*n* = 9). The rats in the control group were fed a control chow diet that followed the AIN-93M formula, with 10% of energy from fat, 15% from protein, and 75% from carbohydrate. The rats in the HFD groups were fed HFD alone or HFD supplemented with 1% or 5% dried onion (DO and 5DO, respectively) or quercetin (Q; 180 mg/kg diet, equal to the quercetin content in 5% dried onion); the HFD formula was modified from the previous study [39], with 43% of energy from fat, 12% from protein, and 45% from carbohydrate (Table 7). The diets were prepared weekly, stored at −20 °C, and thawed to 4 °C every day before being given to the rats. All animals were allowed free access to water during the study. Food intake and body weight were recorded daily. At the end of the experiment, the animals were killed through CO_2_ asphyxiation. Blood, visceral (epididymal, perirenal, and mesenteric), and subcutaneous fat tissues, liver, and eyes were collected. One part of each hepatic lobe, epididymal fat tissue, and right eye were stored in formalin for histological and immunohistological examination, and the remaining samples were stored at −80 °C until analysis.

### 4.3. Fecal Sample Collection and Analysis

Five days before the end of the study, the rats were moved to metabolic cages to collect fecal samples daily, which were then dried at 50 °C and analyzed for their triglyceride, cholesterol, total lipid content, and short-chain fatty acids.

### 4.4. Adipocyte Area in Epididymal Fat

The adipocyte area in epididymal fat was determined using microscopy (Olympus, Tokyo, Japan; 400×) following a method described previously [40]. Images of hematoxylin–eosin (HE)-stained epididymal fat were analyzed using Image J (version 4.0, National Institutes of Health, Bethesda, MD, USA).

### 4.5. Blood Biochemical Analysis

Serum levels of aspartate transaminase (AST), alanine transaminase (ALT), triglycerides, total cholesterol, low-density lipoprotein cholesterol (LDL-C), HDL-C, free fatty acids, and glucose were assessed using an automated clinical chemistry analyzer (Toshiba TBA120 FR, Tokyo, Japan) at Dahua Clinical Analysis Laboratory (Taichung, Taiwan).

### 4.6. Fecal Short-Chain Fatty Acids

Fresh feces samples (0.1 g) were mixed with 3 mL of distilled water and homogenized. After centrifugation (2000× *g*, 25 °C, 5 min), the supernatant was collected; mixed with isocaproic acid (internal standard), 50% H_2_SO_4_ (100 μL), and diethyl ether (1 mL); and then homogenized. Next, short-chain fatty acids in the supernatant was injected onto the gas chromatographic (GC) columns directly and quantified using FID detector (Agilent Technologies, Santa Clara, CA, USA).

### 4.7. Histopathology of the Liver

The rats’ livers were fixed in 10% formalin immediately after removal. For HE staining, 5-μm-thick sections were deparaffinized with xylene, hydrated in a descending series of graded ethanol, stained with hematoxylin for 2 min, rinsed in gentle running water for 2 min, and stained with eosin for 5 s. The slides were examined by a pathologist at 200× magnification. Lesions were evaluated using a histological scoring system for NAFLD proposed by Shackelford et al. [41].

### 4.8. Triglyceride, Total Cholesterol, and Total Lipid Content in the Liver or Feces

Triglyceride and cholesterol content in liver tissues and dried feces was measured using a triglyceride assay kit (Trichylceride-GPO-reagent set; Teco Diagnostics, Anaheim, CA, USA) and a cholesterol assay kit (Cholesterol reagent set; Teco Diagnostics), respectively, in accordance with the instructions. The total lipid content in 30 mg of dried feces was determined using the method described by Tanaka et al. [42]

### 4.9. Proinflammatory Cytokines in the Serum and Liver

TNF-α and IL-6 concentrations in serum and liver tissues were examined using an enzyme-linked immunosorbent assay kit (Bioscience, San Diego, CA, USA).

### 4.10. Thiobarbituric Acid-Reactive Substances (TBARs) Assay

A TBARs assay was performed using the fluorometric method described previously [43]. The tissue sample (0.03 g) used for the assay was first homogenized with 300 μL of lysis buffer (10% Trion-100, 0.1% SDS, and 0.5% sodium deoxycholate) containing 1 mM phenylmethyl sulfonyl fluoride, and it was then centrifuged at 1000× *g* for 30 min at 4 °C. Next, the supernatant was used for the assay. An aliquot of 200 μL of serum sample or tissue lysate was added to 200 μL of trichloroacetic acid buffer, mixed, and centrifuged 5000× *g* at 4 °C for 30 min. Then, 200 μL of the supernatant was taken and 200 μL of thiobarbituric acid buffer was added. After heating at 95 °C for 40 min in the dark, the absorbance was measured using a fluorometer at a wavelength of 532 nm/600 nm. Different concentrations of 1,1,3,3-tetramethoxypropane were used to create a standard curve to calculate the malondialdehyde (MDA) product concentration.

### 4.11. SOD, Catalase, and Glutathione Peroxidase Activity, and Glutathione Level in the Liver

Using the method described by Khan and Ahmed [44], we prepared liver tissue homogenate for determining the activity of SOD, catalase, and glutathione peroxidase as well as the level of glutathione in liver samples. Details of the preparation of tissue homogenate and the methods used to determine these parameters regarding to oxidative status have been described previously [15].

### 4.12. Quantitative Real-Time PCR (qRT-PCR)

For RNA isolation, 30–35 μg of tissue was lysed using 200 µL of TRIzol reagent (Invitrogen, Carlsbad, CA, USA). To the lysate, 0.8 mL of TRIzol was added, and they were combined with 200 μL of chloroform and then incubated for 2 min at room temperature. After centrifugation for 15 min at 8000× *g*, the supernatant was collected, and an equal volume of isopropanol was added to incubate for 10 min, after which the mixture was centrifuged for 10 min at 8000× *g*. To obtain RNA, 500 μL of 75% ethanol was added to the pellet and spun down. This was followed by centrifugation at 8000× *g* for 5 min. An appropriate volume of diethyl pyrocarbonate solution was added to adjust the RNA concentration to approximately 1 µg/µL. The High-Capacity cDNA Reverse Transcription Kit was used for reverse transcription (RT). Briefly, 1–10 ng of total RNA was adjusted to 5 µL and combined with RT Master Mix, which contained 100 mM deoxyribonucleotide triphosphate (with thymidine triphosphate), 50 U/µL MultiScribe Reverse Transcriptase, 20 U/µL RNase inhibitor, and 2.5 M 10X RT primer. When the reaction was complete, the mixture was loaded into the thermal cycler at 25, 37, and 85 °C for 10, 120, and 5 min, respectively, and at 4 °C for cool down. Until RT was complete, the reagents were stored between −15 and −25 °C. Fast SYBR Green Master Mix (Thermo Fisher Scientific, Richmond, VA, USA) was prepared in accordance with the manufacturer’s instructions for quantitative PCR. The primer sequence was as follows: ACC-α, forward primer, 5′-AACATCCCGCACCTTCTTCTAC-3′; reverse primer, 5′-CTTCCACAAACCAGCGTCTC-3′; GAPDH, forward primer, 5′-CAAGTTCAACGGCACAGTCAA-3′; reverse primer, 5′-TGGTGAAGACGCCAGTAGACTC-3′. The mixture was incubated at 95 °C for 5 min, followed by 35–45 cycles at 95, 60, and 72 °C for 5, 10, and 1 s, respectively. For melting curve analysis, the samples were denatured at 95 °C and then cooled to 65 °C at a rate of 20 °C/s. Between 65 and 95 °C, fluorescence signals were continuously obtained at a wavelength of 530 nm at a 0.2 °C/s rate. Gene expression was calculated as ddCT, using the 18S rRNA gene as a reference. Data are expressed relative to the control group normalized to a value of 1. Analysis was performed using a StepOnePlus Real-Time PCR System (Applied Biosystems, Foster City, CA, USA).

### 4.13. Western Blotting

Epididymal fat or eye tissue (0.03 g) was homogenized in 300 μL of phosphate-buffered saline (PBS) containing a protease inhibitor cocktail (Complete, Roche Applied Science, Mannheim, Germany) and a phosphatase inhibitor cocktail (GoalBio, Taipei, Taiwan) with TissuelyserII (QIAGEN, Hilden, Germany). The supernatant was collected to determine the protein concentration and to resolve through Western blotting assay in accordance with the method described previously. Briefly, after the proteins were analyzed through SDS-PAGE and transferred onto a nitrocellulose membrane (Millipore, Bedford, MA, USA), the membrane was blocked with 5% nonfat milk powder in 0.1% Tween-20-Tris-PBS and then incubated overnight with the indicated antibody at 4 °C. Antibodies against acetyl-CoA carboxylase-alpha (ACC-α), phosphoinositide 3-kinases (PI3K), protein kinase B (AKT), RAS, c-RAF, mitogen-activated protein kinase (MEK), p-MEK, and phosphor-extracellular signal-regulated kinase (p-ERK) were purchased from Santa Cruz (Dallas, TX, USA). They were then incubated with antimouse horseradish peroxidase antibodies (GE Healthcare, Buckinghamshire, UK). The signals were developed through chemiluminescence using the Immobilon Western Chemiluminescent HRP Substrate (Merck Millipore, Danvers, MA, USA) and exposure to ECL hyperfilm in the LAS-3000 luminescent image analyzer (Fujifilm, Tokyo, Japan). Protein quantification was performed through densitometry and using the Multi Gauge (version 2.2; Fujifilm, Stockholm, Sweden).

### 4.14. Immunohistochemical Analysis

Formalin-fixed and paraffin-embedded eye sections (5 μm) on coated slides were deparaffinized with xylene and rehydrated in a descending series of graded ethanol. Endogenous peroxidase activity was blocked with 0.6% H_2_O_2_. To show neovascularization, the sections were then incubated with CD31 antibodies (Abcam Co., Cambridge, UK) at a dilution of 1:2000, as suggested by the manufacturers, at 37 °C for 1 h, which was followed by detection with the UltraVision Quanto Detection System HRP and DAB Quanto Chromogen and Substrate (Thermo Fishers Scientific, Waltham, MA, USA), counterstaining with Mayer’s hematoxylin, and mounting in glycerin. The sections were observed at 200× magnification.

### 4.15. Statistical Analysis

Values are expressed as mean ± standard deviation. One-way factorial analysis of variance followed by Duncan’s multiple-range test was used to analyze intragroup differences, and Student’s *t* test was used to evaluate intergroup differences. *p* < 0.05 was considered statistically significant. All statistical analyses of data were performed using SigmaStat 4.0 or SigmaPlot 10.0 (Sigma Sales Solution Co., Delhi, India).

## Figures and Tables

**Figure 1 ijms-23-11091-f001:**
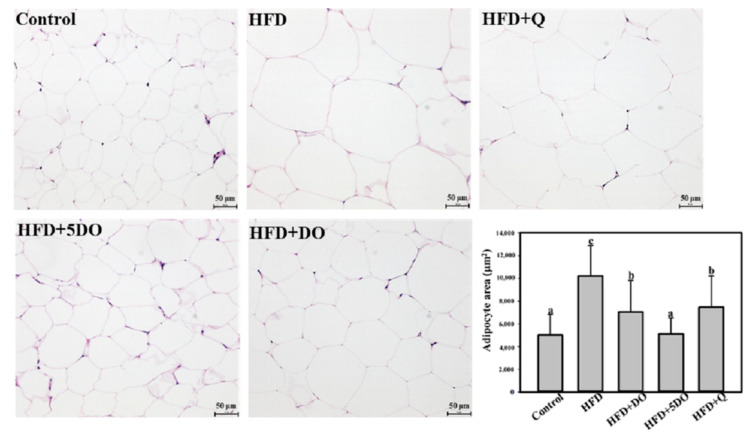
Effect of dried onion powder and quercetin on the HE staining images and the quantitative level of adipocyte area in epididymal adipose in rats exposed to HFD. The animals were treated with a high fat diet (HFD) along or supplemented with dried onion powder (1% or 5%; DO and 5DO, respectively) or quercetin (Q) for 12 weeks. The control group was given the control diet. Values (mean ± SD) not sharing a common letter are significantly different (*p* < 0.05). Scale bar, 50 μm.

**Figure 2 ijms-23-11091-f002:**
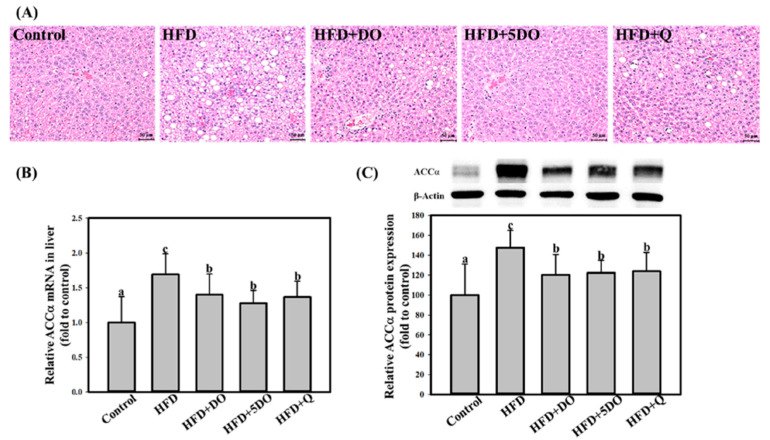
Effect of dried onion powder and quercetin on HE staining images (**A**) as well as mRNA (**B**) and protein (**C**) levels of ACC-α in liver in rats exposed to HFD. The animals were treated with a high fat diet (HFD) along or supplemented with dried onion powder (1% or 5%; DO and 5DO, respectively) or quercetin (Q) for 12 weeks. The control group was given the control diet. Values (mean ± SD) not sharing a common letter are significantly different (*p* < 0.05).

**Figure 3 ijms-23-11091-f003:**
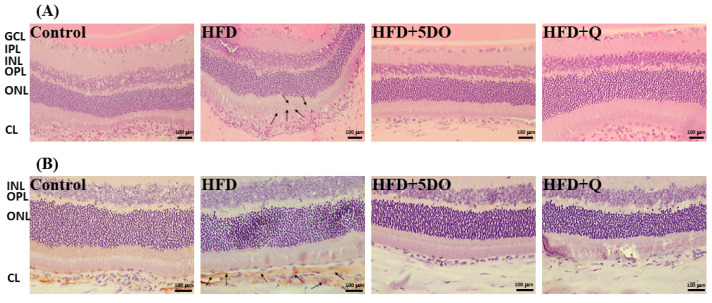
Effect of dried onion powder and quercetin on retinopathy. HE staining images with arrows indicated the adipose-deposition (**A**) and histological images stained with CD-31 antiserum (**B**) in retina of rats exposed to HFD. Scale bar, 100 μm. The CD-31 positive areas were pointed by arrows. The animals were treated with a high fat diet (HFD) along or supplemented with dried onion powder (5%; 5DO) or quercetin (Q) for 12 weeks. The control group was given the control diet. GCL, ganglion cell layer; IPL, inner plexiform layer; INL, inner nuclear layer; OPL, outer plexiform layer; ONL, outer nuclear layer; CL, choroid layer.

**Figure 4 ijms-23-11091-f004:**
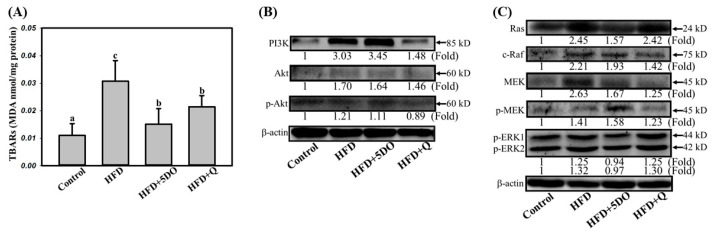
Effect of dried onion powder and quercetin on lipid peroxidation and protein expression in rat retina. Lipid peroxidation (**A**); the expression of PI3K and Akt (**B**) and Ras/Raf/MEK (**C**) in retina of rats exposed to HFD. The retina proteins obtained from animals and analyzed by immunoblot. The protein level was normalized by the control group and represented as fold.

**Figure 5 ijms-23-11091-f005:**
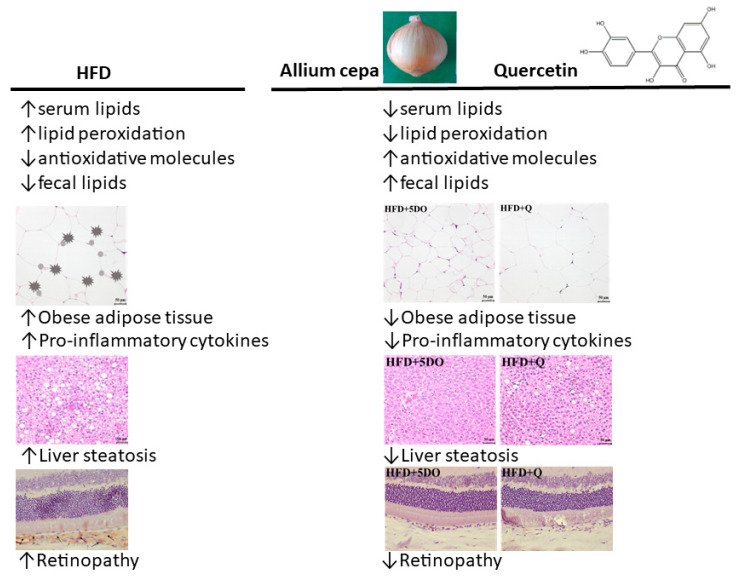
Summary of dried onion powder and quercetin on body lipid, liver steatosis, and retinopathy induced by HFD.

**Table 1 ijms-23-11091-t001:** Effects of dried onion powder and quercetin on food intake, energy intake and body weight in SD rats exposed to HFD ^1^.

	Control	HFD	HFD+ DO	HFD+ 5DO	HFD+ Q
Food intake (g/day)	32.6 ± 1.6 ^b^	23.2 ± 1.6 ^a^	23.1 ± 2.2 ^a^	23.2 ± 1.1 ^a^	22.9 ± 1.8 ^a^
Energy intake (kcal/day)	74.0 ± 4.2 ^a^	90.7 ± 4.4 ^b^	90.7 ± 9.6 ^b^	90.3 ± 4.7 ^b^	89.8 ± 7.9 ^b^
Final body weight (g)	560 ± 55 ^a^	633 ± 32 ^c^	602 ± 40 ^bc^	590 ± 24 ^ab^	620 ± 60 ^bc^

^1^ The animals were treated with a high fat diet (HFD) along or supplemented with dried onion powder (1% or 5%; DO and 5DO, respectively) or quercetin (Q) for 12 weeks. The control group was given the control diet. Values (mean ± SD) not sharing a common letter in the same row are significantly different (*p* < 0.05).

**Table 2 ijms-23-11091-t002:** Effect of dried onion powder and quercetin on body lipid in Sprague Dawley rats exposed to HFD ^1^.

	Control	HFD	HFD + DO	HFD + 5DO	HFD + Q ^2^
Epididymal adipose	26.7 ± 3.3 ^a^	34.2 ± 5.2 ^b^	28.6 ± 6.0 ^a^	27.9 ± 2.6 ^a^	34.2 ± 8.3 ^b^
Perirenal adipose	32.5 ± 5.1 ^a^	44.5 ± 8.7 ^b^	36.0 ± 7.8 ^a^	35.8 ± 6.6 ^a^	38.7 ± 8.5 ^ab^
Mesenteric adipose	16.8 ± 1.9 ^a^	24.0 ± 4.2 ^b^	17.0 ± 4.5 ^a^	14.1 ± 2.0 ^a^	23.1 ± 5.4 ^a^
Subcutaneous adipose	33.8 ± 7.0 ^a^	49.7 ± 4.3 ^c^	45.8 ± 10.0 ^bc^	40.9 ± 6.3 ^ab^	48.1 ± 13.4 ^c^

^1^ The animals were treated with a high fat diet (HFD) along or supplemented with dried onion powder (1% or 5%; DO and 5DO, respectively) or quercetin (Q) for 12 weeks. The control group was given the control diet. Values (means ± SD, *n* = 9–10) not sharing a common letter in the same row are significantly different (*p* < 0.05). ^2^ mg/g body weight.

**Table 3 ijms-23-11091-t003:** Effects of dried onion powder and quercetin on the blood biochemistry values, pro-inflammatory cytokines and TBARs in Sprague Dawley rats exposed to HFD ^1^.

	Control	HFD	HFD + DO	HFD + 5DO	HFD + Q
ALT (IU/L)	30 ± 6 ^a^	39 ± 12 ^b^	30 ± 4 ^a^	30 ± 6 ^a^	36 ± 7 ^a^
AST (IU/L)	83 ± 13 ^a^	114 ± 35 ^b^	86 ± 11 ^a^	81 ± 7 ^a^	93 ± 19 ^a^
Triglycerides (mg/dL)	104 ± 23 ^c^	87 ± 15 ^bc^	74 ± 23 ^ab^	58 ± 9 ^a^	80 ± 25 ^b^
Total cholesterol (mg/dL)	70.9 ± 8.5 ^b^	64.1 ± 7.6 ^ab^	61.5 ± 8.6 ^a^	57.7 ± 7.9 ^a^	63.5 ± 10.4 ^ab^
LDL-cholesterol (mg/dL)	23.5 ± 2.6 ^c^	21.8 ± 3.4 ^bc^	19.0 ± 3.0 ^ab^	18.3 ± 4.4 ^a^	21.1 ± 1.4 ^abc^
HDL-cholesterol (mg/dL)	27.1 ± 1.5 ^b^	23.6 ± 2.2 ^a^	23.3 ± 1.7 ^a^	24.8 ± 2.2 ^a^	24.0 ± 1.9 ^a^
Free fatty acid (mg/dL)	0.51 ± 0.03 ^ab^	0.52 ± 0.03 ^b^	0.46 ± 0.02 ^a^	0.45 ± 0.01 ^a^	0.50 ± 0.04 ^a^
Glucose (mg/dL)	153 ± 14 ^ab^	166 ± 21 ^b^	164 ± 18 ^b^	142 ± 20 ^a^	169 ± 14 ^b^
IL-6 (pg/mL)	0.9 ± 0.4 ^a^	1.2 ± 0.5 ^b^	0.9 ± 0.3 ^a^	0.7 ± 0.3 a	0.9 ± 0.3 a
TNF-α (pg/mL)	10.0 ± 1.8 ^a^	15.8 ± 6.7 ^b^	14.5 ± 4.8 ^bc^	10.9 ± 3.4 ^ab^	11.6 ± 2.2 ^ab^
TBARs (MDA nmole/mL)	0.9.0 ± 0.5 ^a^	1.3 ± 0.3 ^b^	0.9 ± 0.4 ^a^	0.9 ± 0.3 ^a^	1.0 ± 0.4 ^ab^

^1^ The animals were treated with a high fat diet (HFD) along or supplemented with dried onion powder (1% or 5%; DO and 5DO, respectively) or quercetin (Q) for 12 weeks. The control group was given the control diet. Values (means ± SD, *n* = 9–10) not sharing a common letter in the same row are significantly different (*p* < 0.05).

**Table 4 ijms-23-11091-t004:** Effects of dried onion powder and quercetin on lipid profile and the levels of pro-inflammatory cytokine and anti-oxidative parameters in liver in Sprague Dawley rats exposed to HFD ^1^.

	Control	HFD	HFD + DO	HFD + 5DO	HFD + Q
Triglycerides (mg/g protein)	128 ± 16 ^b^	164 ± 20 ^c^	101 ± 17 ^a^	115 ± 20 ^ab^	150 ± 9 ^c^
Total Cholesterol (mg/g protein)	134 ± 15 ^a^	192 ± 14 ^d^	160 ± 8 ^c^	139 ± 7 ^ab^	149 ± 7 ^b^
IL-6 (pg/mL)	479 ± 92 ^c^	512 ± 109 ^c^	378 ± 128 ^ab^	302 ± 105 ^a^	428 ± 140 ^bc^
TNF-α (pg/mL)	99 ± 21 ^ab^	106 ± 17 ^b^	95 ± 22 ^ab^	86 ± 14 ^a^	99 ± 20 ^ab^
SOD (unit/mg protein)	0.12 ± 0.05 ^ab^	0.11 ± 0.02 ^ab^	0.13 ± 0.03 ^ab^	0.14 ± 0.03 ^b^	0.11 ± 0.02 ^a^
Catalase (unit/mg protein)	6.1 ± 1.9 ^c^	3.5 ± 2.2 ^ab^	4.7 ± 2.0 ^bc^	6.11 ± 1.23 ^c^	2.7 ± 1.3 ^a^
Gpx (nmole NADPH oxidized/min/mg protein)	16.9 ± 2.2 ^cd^	9.0 ± 1.5 ^a^	14.9 ± 1.1 ^c^	17.7 ± 4.0 ^d^	12.3 ± 2.3 ^b^
Glutathione (nmole/mg protein)	14.0 ± 3.9 ^b^	9.0 ± 2.5 ^a^	13.1 ± 3.0 ^b^	13.7 ± 3.6 ^b^	11.8 ± 2.6 ^ab^
TBARs (MDA nmole/mg protein)	0.12 ± 0.03 ^a^	0.31 ± 0.14 ^b^	0.20 ± 0.04 ^a^	0.19 ± 0.07 ^a^	0.23 ± 0.04 ^ab^

^1^ The animals were treated with a high fat diet (HFD) along or supplemented with dried onion powder (1% or 5%; DO and 5DO, respectively) or quercetin (Q) for 12 weeks. The control group was given the control diet. Values (means ± SD, *n* = 9–10) not sharing a common letter in the same row are significantly different (*p* < 0.05).

**Table 5 ijms-23-11091-t005:** Dried onion powder and quercetin reduced hepatic steatosis in Sprague Dawley rats exposed to HFD ^1^.

	Control	HFD	HFD + DO	HFD + 5DO	HFD + Q ^2^
Fatty change, general	2.25 ± 1.28	3.13 ± 1.13	2.44 ± 1.24	2.33 ± 0.71 ^#^	2.56 ± 1.01
Inflammation, multifocal	0.63 ± 0.52	1.25 ± 0.71 *	0.78 ± 0.44	0.33 ± 0.50 ^#^	0.78 ± 0.44
Fatty change with macro-vesicles, multifocal	1.25 ± 0.71	1.88 ± 1.13	1.78 ± 0.83	1.00 ± 0.71 ^#^	1.33 ± 0.50
Fatty change with micro-vesicles, multifocal	2.25 ± 1.28	3.13 ± 1.13	2.44 ± 1.24	1.89 ± 1.05 ^#^	2.33 ± 0.71 ^#^

^1^ The animals were treated with a high fat diet (HFD) along or supplemented with dried onion powder (1% or 5%; DO and 5DO, respectively) or quercetin (Q) for 12 weeks. The control group was given the control diet. Values (mean ± SD) with a * denote *p* < 0.05 versus control and a ^#^ denote *p* < 0.05 versus HFD. ^2^ histopathological score.

**Table 6 ijms-23-11091-t006:** Effects of dried onion powder and quercetin on fecal lipid profiles in Sprague Dawley rats exposed to HFD ^1^.

	Control	HFD	HFD + DO	HFD + 5DO	HFD + Q
Total lipid (mg/g dry wt)	178 ± 13 ^a^	173 ± 9 ^a^	168 ± 17 ^a^	190 ± 13 ^b^	168 ± 16 ^a^
Triglycerides (mg/g dry wt)	1.0 ± 0.5 ^a^	2.2 ± 1.1 ^b^	3.9 ± 0.5 ^c^	4.1 ± 0.5 ^c^	0.9 ± 0.1 ^a^
Total cholesterol (mg/g dry wt)	1.0 ± 0.4 ^a^	2.0 ± 0.3 ^b^	2.1 ± 0.2 ^b^	2.4 ± 0.3 ^c^	0.8 ± 0.2 ^a^
Formic acid (μmol/g dry wt)	8.4 ± 9.9 ^a^	8.9 ± 8.5 ^a^	8.6 ± 8.5 ^a^	2.3 ± 2.2 ^a^	11.6 ± 8.4 ^a^
Acetic acid (μmol/g dry wt)	3.5 ± 1.6 ^a^	4.3 ± 1.1 ^ab^	4.0 ± 1.5 ^ab^	5.2 ± 0.3 ^b^	4.9 ± 0.8 ^b^
Isobutyric acid (μmol/g dry wt)	3.6 ± 0.1 ^a^	3.6 ± 0.2 ^a^	3.7 ± 0.2 ^ab^	3.8 ± 0.04 ^b^	3.59 ± 0.1 ^a^
Butyric acid (μmol/g dry wt)	1.1 ± 0.7 ^a^	1.4 ± 0.8 ^ab^	1.3 ± 0.5 ^ab^	2.1 ± 0.9 ^b^	2.2 ± 1.0 ^b^

^1^ The animals were treated with a high fat diet (HFD) along or supplemented with dried onion powder (1% or 5%; DO and 5DO, respectively) or quercetin (Q) for 12 weeks. The control group was given the control diet. Values (means ± SD, *n* = 9–10) not sharing a common letter in the same row are significantly different (*p* < 0.05).

**Table 7 ijms-23-11091-t007:** Composition of experimental diets.

	Group ^1^	Control	HFD	HFD+ DO	HFD + 5DO	HFD+ Q
Ingredient (g/kg)						
Cornstarch		466	276	267	230	280
Dextrinized cornstarch		155	155	155	155	155
Casein		140	140	140	140	140
Sucrose		100	100	100	100	100
Soybean oil		40	40	40	40	40
Lard		0	190	190	190	190
Fiber		50	50	49.1	45.6	50
Mineral mix ^2^		35	35	35	35	35
Vitamin mix ^3^		10	10	10	10	10
Cysteine		1.8	1.8	1.8	1.8	1.8
Choline bitartrate		2.5	3	3	3	3
Tert-butylhydroquinone		0.01	0.01	0.01	0.01	0.01
Dried onion powder		-	-	10	50	-
Quercetin		-	-	-	-	0.18

^1^ The animals were treated with a high fat diet (HFD) or the HFD supplemented with dried onion powder (1% or 5%; DO and 5DO, respectively) or quercetin (Q) for 12 weeks. The control group was given the control diet. ^2^ AIN-93M-MX mineral mixture. ^3^ AIN-93-VX vitamin mix.

## Data Availability

Not applicable.

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
