# Peer review of "Effects of Dried Onion Powder and Quercetin on Obesity-Associated Hepatic Menifestation and Retinopathy"

_ijms, 2022, doi:10.3390/ijms231911091_

Round 1
Reviewer 1 Report
Please refer to attached file.

Author Response
Response to Reviewer 1
The paper is generally well written with a minimum of typographical errors and the subject of the study is interesting.
Response
We appreciate your positive comments.
General comments and questions:
- As onion is often consumed in its cooked form, does the claimed health benefits of onion also apply to cooked onion?
Response
We did not determine the effects of cooked onion in the current study. Recent clinical studies showed that daily intake of quercetin-rich onion powder (sterilized at 60 â—¦C for 120 min) for 12 weeks may be beneficial for preventing obesity and improving liver function in healthy Japanese subjects (Nishimura et al. (2019); steamed onion powder (99â—¦C for 4 h) supplementation reduces body fat as well as serum triglyceride and C-peptide in overweight subjects in South Korea (Jeong et al.,2020). These studies suggest that cooked onion powder may still possess bioactivities. However, further studies are needed to confirm this issue.
Jeong S, Chae J, Lee G, Shin G, Kwon YI, Oh JB, Shin DY, Lee JH. Effect of Steamed Onion (ONIRO) Consumption on Body Fat and Metabolic Profiles in Overweight Subjects: A 12-Week Randomized, Double-Blind, Placebo-Controlled Clinical Trial. J Am Coll Nutr. 2020 Mar-Apr;39(3):206-215.
Nishimura M, Muro T, Kobori M, Nishihira J. Effect of Daily Ingestion of Quercetin-Rich Onion Powder for 12 Weeks on Visceral Fat: A Randomised, Double-Blind, Placebo-Controlled, Parallel-Group Study. Nutrients. 2019 Dec 28;12(1):91.
- 2. In the Introduction section, paragraph 3, “A diet containing 5% onion…” is it 5% w/w or w/v?
Response
Thank you for the reminder, A diet containing 5% onion (w/w) has been revised.
In section 2.1:
- 3. In the Tables, the term ‘SD rats’ is used – presumably this refers to ‘Sprague Dawley rats’ – this should be mentioned in the text or table legend. The abbreviation ‘SD’ is also used for ‘standard deviation’ – perhaps the authors could use ‘sd’ instead.
Response
Thank you for the reminder. To avoid the misunderstanding, the “SD rats” in all table heads has been revised to the full name “Sprague Dawley rats”.
- 4. The term “…lower food intake volume (Table 1).” should read as “…lower food intake mass (Table 1).”?
Response
Thank you. It has been revised.
- 5. The number of rat replicates should be mentioned before the presentation of results.
Response
Thank you. It has been described page 2, section 2.1.
- 6. The title of Table 2 is confusing – the title mentions “…reduced body lipid…” but most of the results show increased values.
Response
Thank you for the reminder. The head of Table has been revised to “Effect of dried onion powder and quercetin on body lipid in Sprague Dawley rats exposed to HFD”.
- 7. ALT, AST and TBARs should be explained earlier in the text or as legend to Table 3.
Response
Thank you. It has been revised as “HFD significantly increased the levels of aspartate transaminase (AST), alanine transaminase (ALT), ALT, AST, interleukin (IL)-6, tumor necrosis factor (TNF)-α, and oxidative stress (TBARs) and decreased the HDL-C level (Table 3). It revealed that HFD impaired the liver function, increased inflammatory cytokines and oxidative stress.”, in section 2.2.
- 8. In section 4.1 the authors mention that the onions are homogenized to a fine powder – does this mean that dried onions were used? If so, how were the onions dried?
Response
Yes, the dried onion was used in this study. To make the preparation procedure read clearly, the relative description in section 4.1 has been revised to “After peeling off their skins, the onions were cut into small pieces. Next, the onion pieces were lyophilize at −40 °C for 24 hrs (Chaosu Company, Minhsiung Township, Chiayi County, Taiwan) then mechanically gridded into fine powder, and stored at −20 °C until use.”
- 9. Section 4.1 mentions the analysis of quercetin. However, there are other bioactive compounds like polyphenols anthocyanins, and flavonoids that can play a role (besides fiber). It would have been preferable if the chemical characterization extended beyond quercetin so that future studies could compare chemical profiles for product consistency.
Response
Thank you for the kind comment. We agree the opinion of the reviewer. And as suggested, we will compare chemical profiles for product consistency in future studies.
- 10. Insoluble fiber was determined but why was soluble fiber not also determined as its importance was acknowledged in the second paragraph of the Discussion section.
Response
Thank you for the reminder. We now have added the information about soluble fiber (total fructan) content to the Materials and Methods (section 4.1), as “… and total fructan content was determined by AOAC 999.03-2005; 100 g of the DO contained 8.86 g crude fiber (insoluble fiber) and 3.23 g fructans (soluble fiber).”
- 11. In section 4.6, was the acids quantified by GC directly or following derivatization?
Response
Short-chain fatty acids in the supernatant were injected onto the gas chromatographic (GC) columns directly and quantified using FID detector. We have now modified the description as “… the supernatant was injected onto the gas chromatographic (GC) columns directly and quantified…”.
- 12. In section 4.8 “… using the method of…” what?
Response
Thank you. It has been revised to “…using the method described by Tanaka et al.”
- 13. (1) Ideally the authors should have also examined the fecal microbiota due to its importance as mentioned in the last paragraph of the Introduction. (2) In the Discussion section, second paragraph, it is mentioned that “…, we calculated the fiber volume in the OD and 5OD groups and added proper fiber to the other groups (Table 7).” – does this include both soluble and insoluble fiber? Both types of fiber should be included.
Response
- We agreed the opinion of reviewer that we should examine the fecal microbiota. We will perform the examination in our future study.
- As mentioned above (Q 10), we have determined the mass of soluble fiber and added the information in the Materials and Methods (section 4.1). Because only insoluble fiber was added to the diets in the current study, we have modified the description in the Discussion (the 2nd paragraph) as the following, “We speculated that the soluble fiber (such as fructans that is, inulin and fructooligo-saccharides) in DO/5DO may contribute to the increased fecal lipid excretion, because insoluble fiber was supplemented to the diets in each group (Table 7). Hiel et al. [23] demonstrated that inulin supplementation improves postprandial hypertriglyceridemia by modulating gene expression and increases fecal lipid excretion.”
Reviewer 2 Report
The research article "Effects of dried onion powder and quercetin on obesity-associated hepatic manifestations and retinopathy" is a good in vitro and in vivo study.
• This article will help the scientific community involved in diabetes, cardiovascular disease, cancer, and nonalcoholic fatty liver disease
The team studied: The effects of dried onion powder (DO) on a high-fat diet-induced disorder in VFA accumulation, liver health, and retinal neovascularization in rats and compared it with quercetin-flavonoid.
The essential aspects of the article follow:
• DO can reduce HFD-induced body weight gain, fat, and hepatic lipid accumulation.
• DO enhanced fecal fat, thus reductions in lipid content.
• DO increased hepatic antioxidant enzyme activity and glutathione levels and had anti-inflammatory/antioxidant properties.
• DO, and quercetin individually have benefits for reducing body lipids, liver fat, oxidative stress, and inflammatory cytokines.
• DO or quercetin reduced retinal adiposity and neovascularization
• DO or quercetin protects from HFD-induced retinal damage.
Questions:
- Explain why CD 31 antibody was used? Explain why targeting endothelial cell marker.
- Name the retinal tissue section layer (fig.3), and also include the scale bar
- Why the retinal tissue section layer seems different in Fig.3 (Row A in the HFD)? Is there any difference in the tissue morphology in the different conditions?
Author Response
Response to Reviewer 2
- Explain why CD 31 antibody was used? Explain why targeting endothelial cell marker.
Response
CD31 antibody is an endothelial cell marker to reveal the neovascularization. On the basis, eye section was examined the expression of CD31 to evaluate the neovascularization. The rationale has been indicated in section 2.5.
- Name the retinal tissue section layer (fig.3), and also include the scale bar.
Response
Thank you. The retinal tissue section layers have been named and the scale bar has been added in Figure 3 along with the revised legend.
- Why the retinal tissue section layer seems different in Fig.3 (Row A in the HFD)? Is there any difference in the tissue morphology in the different conditions?
Response
Thank you for the question. In Figure 3, the bow-shaped section is often appeared in histological examination of eye retina due to the whole eye specimen embedded. In the current study, the tissue morphology did not show differences but lipid deposited within choroid layer.
Reviewer 3 Report
This is a very interesting work done by the authors. Kindly note that since there were no line numbers on the draft, all comments will reflect the quoted text.
"Dietary habits and lifestyle influence the prevalence of overweight and obesity" Kindly include relevant reference to support this statement.
"the health burden of these conditions is increasing worldwide, including in Taiwan, and they are causally linked with the development of several chronic diseases such as diabetes, cardiovascular disease, cancer, and nonalcoholic fatty liver disease (NAFLD)" Please consider if the reference provided is a key reference to support this statement and revise appropriately.
"Overweight and obesity are symptoms of abnormal or excessive visceral and subcutaneous fat accumulation." Perhaps the authors would like to consider citing WHO on this definition (or any other source that reflect it).
"hypertrophic adipocytes secrete proinflammatory cytokines, such as interleukin (IL)-6 and tumor necrosis factor (TNF)-α, into circulation and increase inflammation in various organs". Please include relevant reference to support your statement.
"Onion (Allium cepa L.) is a vegetable commonly consumed worldwide and contains diverse phytochemicals, including organosulfur compounds, phenolic compounds, poly-saccharides, and saponins." Kindly include relevant reference to support this statement. Also, since the object of investigation is dried onion, perhaps in this paragraph the authors would consider including some key remarks about the differences of this particular food. (why is it worth looking into this, aside from the fact that noone else did? and why dring the onion is expected to be useful in terms of bioactive potential? - condensed bioactive power? reduction of required amount of consumption? etc) it is essential for the reader to understand that this research's planing - and choice of the item to be studied - aimed to fill the gap of current knowledge and introduce something to be studied further. Please consider this and rephrase appropreately.
"We also compared the effects of DO with those of quercetin, a major flavonoid present in onions" In the same way as the previous comment, perhaps the authors would like to consider explaining in brief why the comparison between a pure compound and a complex food is useful. There is relevant research to support that co-existing compounds in several foods construct a more effective biological action that the isolated forms (same goes for several highly active components of foods that work better alone). In this context, perhaps it would be useful for the reader to have a clear view of this background information, to better understand the comparison. Please consider and revise appropriately.
"demonstrated that inulin supplementation improves postprandial hypertriglyceridemia by modulating gene expression and increases fecal lipid excretion." Somethig seems to be missing here. Please revisit.
"The soluble fibers present in DO, such as inulin and fructo-oligosaccharides" Please include relevant reference to support this statement. Since this current work doesn't target compound identification on the samples of DO prepared for the research protocol, perhaps the authors would like to consider including previous research that has identified such items and hence we now know they are there.
"Soluble fibers, also known as prebiotics, stimulate the growth of probiotics and production of short-chain fatty acids." Please include relevant reference.
"Onion samples were obtained" Please include in brief if those were raw products.
"homogenized into a fine powder. Next, the onion powder was freeze-dried at −40 °C, mechanically crushed into fine powder," Please consider rephrasing so that the reader will have a clear understading in which step the fine powder was made.
"This finding agrees with that of" Somethig seems to be missing here. Please revisit.
Please mind the overall format of the text - there seems to be somethign defferent toward the end of discussion.
Kindly revisit the format of the references at the end to make sure they are all presented in the same way (there are items that seem to be missing in some). Also please consider revisiting the references dated before the last two decades and perhaps include updated insights that serve to the presentation of this work.
Author Response
Response to Reviewer 3
This is a very interesting work done by the authors. Kindly note that since there were no line numbers on the draft, all comments will reflect the quoted text.
Response
Thank you for the positive comments. The line number has been added on the right side in the text.
- "Dietary habits and lifestyle influence the prevalence of overweight and obesity" Kindly include relevant reference to support this statement.
Response
Thank you. We have added a reference (renewal #1, in Reference) regarding to this statement.
- "the health burden of these conditions is increasing worldwide, including in Taiwan, and they are causally linked with the development of several chronic diseases such as diabetes, cardiovascular disease, cancer, and nonalcoholic fatty liver disease (NAFLD)" Please consider if the reference provided is a key reference to support this statement and revise appropriately.
Response
Thank you. We have renewed the cited literature as updated #2, in Reference.
- "Overweight and obesity are symptoms of abnormal or excessive visceral and subcutaneous fat accumulation." Perhaps the authors would like to consider citing WHO on this definition (or any other source that reflect it).
Response
Thank you. The description is referred to WHO definition and has been modified to “Overweight and obesity display abnormal or excessive fat accumulation, including visceral and subcutaneous fat.”, along with an additional literature (renewal #3, in Reference)
- "hypertrophic adipocytes secrete proinflammatory cytokines, such as interleukin (IL)-6 and tumor necrosis factor (TNF)-α, into circulation and increase inflammation in various organs". Please include relevant reference to support your statement.
Response
Thank you. We have added a reference to support the statement (renewal #7, in Reference).
- "Onion (Allium cepa L.) is a vegetable commonly consumed worldwide and contains diverse phytochemicals, including organosulfur compounds, phenolic compounds, poly-saccharides, and saponins." Kindly include relevant reference to support this statement. Also, since the object of investigation is dried onion, perhaps in this paragraph the authors would consider including some key remarks about the differences of this particular food. (why is it worth looking into this, aside from the fact that noone else did? and why dring the onion is expected to be useful in terms of bioactive potential? - condensed bioactive power? reduction of required amount of consumption? etc) it is essential for the reader to understand that this research's planing - and choice of the item to be studied - aimed to fill the gap of current knowledge and introduce something to be studied further. Please consider this and rephrase appropreately.
Response
(1) As suggested, we have added the reference to support this statement (#10 in Reference). In fact, we had cited it at the end of next sentence.
(2) We have now added more information about why we investigated freeze-dried onion powder as the followings (the last paragraph in the Introduction). The additional reference has been added in Reference, #16.
“Because DO is reportedly shown that the contents of flavonoid (90% are quercetin and quercetin derivatives) in black onion and the indicated flavonoids are decreased during heat treatment [16], we used DO, which was prepared by freeze-drying at -40℃, to perform the current study. Beside, DO is a kind of condensed bioactive powder, which reduces the required amount of consumption for the health benefits of onion and may facilitate the development of onion as a functional food for preventing obesity and the associated healthy issues.”
- "We also compared the effects of DO with those of quercetin, a major flavonoid present in onions" In the same way as the previous comment, perhaps the authors would like to consider explaining in brief why the comparison between a pure compound and a complex food is useful. There is relevant research to support that co-existing compounds in several foods construct a more effective biological action that the isolated forms (same goes for several highly active components of foods that work better alone). In this context, perhaps it would be useful for the reader to have a clear view of this background information, to better understand the comparison. Please consider and revise appropriately.
Response
Thank the reviewer for the constructive suggestion. We have now added a brief explanation (as follows) to the last paragraph in the Introduction too.
“Moreover, it has been indicated that quercetin, a major flavonoid present in onions [17] can protect mice against HFD-induced NAFLD by modulating intestinal microbiota im-balance and related gut–liver axis activation [18]. However, whether those co-existing compounds, such as fibers, in onion affect such effects of quercetin is unclear, thus se also compared the effects of DO with those of quercetin alone.”
- "demonstrated that inulin supplementation improves postprandial hypertriglyceridemia by modulating gene expression and increases fecal lipid excretion." Somethig seems to be missing here. Please revisit.
Response
Thank you. The full sentence has been done as “Hiel et al. [23] demonstrated that inulin supplementation improves postprandial hy-pertriglyceridemia by modulating gene expression and increases fecal lipid excretion.”
- "The soluble fibers present in DO, such as inulin and fructo-oligosaccharides" Please include relevant reference to support this statement. Since this current work doesn't target compound identification on the samples of DO prepared for the research protocol, perhaps the authors would like to consider including previous research that has identified such items and hence we now know they are there.
Response
Thank you for the comment. We have determined the mass of soluble fiber and added the information about soluble fiber (total fructan) content to the Materials and Methods (section 4.1), as “… and total fructan content was determined by AOAC 999.03-2005; 100 g of the DO contained 8.86 g crude fiber (insoluble fiber) and 3.23 g fructans (soluble fiber).”
We also modified the description in the Discussion (the 2nd paragraph) as the following, “We speculated that the soluble fiber (such as fructans that is, inulin and fructooligosaccharides) in DO/5DO may contribute to the increased fecal lipid excretion, because insoluble fiber was supplemented to the diets in each group (Table 7). Hiel et al. [23] demonstrated that inulin supplementation improves postprandial hypertriglyceridemia by modulating gene expression and increases fecal lipid excretion.”
- "Soluble fibers, also known as prebiotics, stimulate the growth of probiotics and production of short-chain fatty acids." Please include relevant reference.
Response
Thank you. The reference has been added in renewal Reference, #24.
- "Onion samples were obtained" Please include in brief if those were raw products.
Response
Thank you. The description has been revised to “Fresh onion were obtained...”
- "homogenized into a fine powder. Next, the onion powder was freeze-dried at −40 °C, mechanically crushed into fine powder," Please consider rephrasing so that the reader will have a clear understanding in which step the fine powder was made.
Response
Thank you for the reminder. To make the reader understand, the procedures have been modified to “After peeling off their skins, the onions were cut into small pieces. Next, the onion pieces were lyophilized at −40 °C for 24 hrs (Chaosu Company, Minhsiung Township, Chiayi County, Taiwan) then gridded into fine powder…”
- "This finding agrees with that of" Something seems to be missing here. Please revisit.
Response
Thank you. The sentence has been revised to “This finding agrees with that the study of Porras et al.”
- Please mind the overall format of the text - there seems to be somethign defferent toward the end of discussion.
Response
Thank you for the kind reminder. We have reviewed the overall format of the text to make it consistent.
- Kindly revisit the format of the references at the end to make sure they are all presented in the same way (there are items that seem to be missing in some). Also please consider revisiting the references dated before the last two decades and perhaps include updated insights that serve to the presentation of this work.
Response
Thank you for the suggestion. We have re-done the reference format to present it in same way. Regarding to the references dated before the last two decades, we have updated the reference, in renewal Reference, #42.